**Estimating the sensitivity of the Priestley-Taylor coefficient to air**
**temperature and humidity**
Ziwei Liu, Hanbo Yang *, Changming Li, Taihua Wang
State Key Laboratory of Hydro-science and Engineering, Department of Hydraulic
Engineering, Tsinghua University, Beijing, China
*Correspondence to*: Hanbo Yang (yanghanbo@tsinghua.edu.cn)

## Abstract

Priestley-Taylor (PT) coefficient (α) is generally set as a constant value or fitted as an empirical function of environmental variables, and it can bias the evaporation estimation or hydrological projections under global warming. By using an atmospheric boundary layer model, this study derives a theoretical and parameter-free equation for estimating α as a function of air temperature (T) and specific humidity (Q). With observations from several water bodies and non-water-limited land sites, we demonstrate that in addition to well estimating the value of α, the derived expressions can also capture the sensitivity of α to T and Q, that is, dα/dT and dα/dQ. α is generally negatively associated with T and Q, of which T plays a more fundamental role in controlling α behaviors. Based on climate model data, we further show that this negative relationship between α and T is of great importance for long-term hydrological predictions. We also provide a lookup graph for practical and broad uses to directly find the values of dα/dT and dα/dQ under specific conditions. Overall, the derived expression gives a physically clear and straightforward approach to quantify changes in α, which is essential for PT-based hydrological simulation and projections.

## 1. Introduction

Evaporation from wet surfaces, including oceans, lakes, and reservoirs, is relevant to global hydrological cycles and water availability. There is a long history of developing theories and methods to estimate wet surface evaporation (Bowen, 1926; Penman, 1948; Priestley and Taylor, 1972; Thornthwaite and Holzman, 1939; Yang and Roderick, 2019). Among existing models, the Priestley-Taylor (PT) model/equation is known for its transparent structure and low input requirement (Priestley and Taylor, 1972). The PT equation is widely used in evaporation estimation across varied scales and is the basis for various hydrologic and land surface models. Specifically, the PT equation comes from the equilibrium evaporation ($\lambda E_{eq}$), and $\lambda E_{eq}$ can be calculated as (Slatyer and Mcilroy, 1961):

$$\lambda E_{eq} = \frac{\varepsilon_a}{\varepsilon_a + 1}(R_n - G) \tag{1}$$

where $\lambda$ (J/kg) is the latent heat of water vaporization, $\varepsilon_a = \Delta/\gamma$, $\Delta$ (kPa/K) is the slope of the saturated vapor pressure versus temperature curve (a function of temperature), and $\gamma$ is the psychrometric constant. $\varepsilon_a$ is a function of air temperature (T). $R_n$-G (kPa/K) is the available energy. The equilibrium evaporation indicates that the near-surface air is saturated, supposing the vapor pressure deficit (VPD) is zero. However, it does not exist in the real world (Brutsaert and Stricker, 1979; Lhomme, 1997a), due to the continuous exchanges of warm and dry air from the entrainment layer, although water is continuously transported from the bottom wet surface into the atmosphere through the evaporation process (Figure 1).

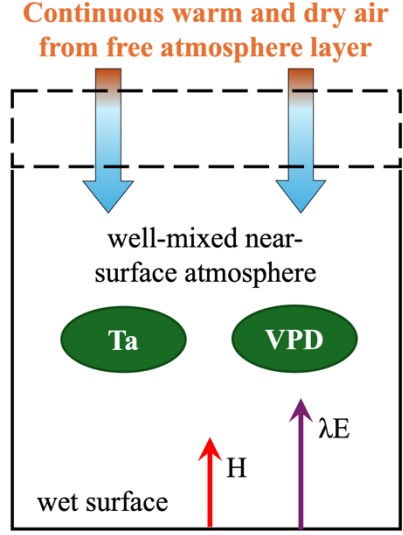

Figure 1. Atmospheric boundary layer box model describing the energy and water fluxes at the saturated surface and atmosphere above. The dotted line represents the removable upper boundary of the box. H and λE are the sensible and latent heat fluxes. Ta is the air

temperature and VPD is the vapor pressure deficit.

In this case, the PT equation introduced a parameter, α, known as the PT coefficient, to
estimate wet surface evaporation (Priestley and Taylor, 1972). α represents the effects of
vertical mixing of dry and moist air and adjusts the equilibrium evaporation to the actual
evaporation. So qualitatively speaking, the α is impossibly lower than one because the air
is always not statured and can only infinitely close to saturated condition, no matter how
moist the near-surface air is. The PT equation is:
$$\lambda E = \alpha \frac{\varepsilon_a}{\varepsilon_a + 1}(R_n - G) \tag{2}$$

In the original study of Priestley and Taylor (1972), the value of α is fitted as 1.26. While
a fixed α value can reasonably estimate wet surface evaporation (Yang and Roderick,
2019), some studies found that α varies across time and space, for example, α often shows
a more prominent value under cold conditions and becomes lower as warms (Xiao et al.,
2020; Debruin and Keijman, 1979). This indicates that α should be a variable rather than
a constant (Assouline et al., 2016; Guo et al., 2015; Jury and Tanner, 1975; Lhomme,
1997b; Van Heerwaarden et al., 2009; Eichinger et al., 1996; Mcnaughton and Spriggs,
1986; Crago et al., 2023; Maes et al., 2019). However, the hydrology field predominantly
employs the fixed value of α = 1.26, despite those earlier findings being over three
decades old.
A general method to quantify the changes in α is to inverse it with observations based on
Equation (2) and then build relationships among α and investigated variables. Since a
negative relationship between α and temperature (T) is a consensus from multi-scale
observations (Assouline et al., 2016; Xiao et al., 2020), many attempts empirically fitted
α as a function of T (Andreas and Cash, 1996; Hicks and Hess, 1977; Yang and Roderick,
2019). Recent work further showed that the air humidity state can also influence the
spatiotemporal patterns of α (Su and Singh, 2023). While those methods promote our
understanding of the potential variations in α, they more lie on the empirical side and pay
less attention to the underlying process. Hence, various endeavors have been made to
calculate α through physical means, but they are often constrained by the complexity of
numerous parameters. For instance, in the research conducted by Lhomme (1997b), α was
explicitly formulated utilizing the PM model in conjunction with boundary layer theory.
Nevertheless, the formulation incorporates parameters that signify surface and
aerodynamic resistances, making them hard to determine through direct measurements.
Subsequently, by using a refined boundary layer model, Van Heerwaarden et al. (2009)
introduced a mathematical expression for estimating α, however, the expression also
involves a set of parameters necessitating numerical experiments to delineate a feasible
range for α. Consequently, obtaining a precise α estimation using conventional
observations still has remained a challenge.
Based on a recent study by Liu and Yang (2021), here we aim to derive a physically clear,
transparent, and calibration-free equation for estimating α, by introducing a governing
equation (potential vapor pressure deficit budget) into the conventional boundary layer
model. In the following sections, we will first provide the theory for estimating α and its
sensitivity to climate conditions: air temperature (T) and humidity (represented by the air
specific humidity, Q). We further evaluate the theory based on measurements from the
water and non-water-limited land surfaces, followed by the influences of α changes on
long-term hydrologic projections.
**2. Theory**
**2.1 Derivation of Bowen ratio**
Here, we use an atmospheric boundary layer-based (ABL) model as the basis for the
Bowen ratio (defined as the ratio of sensible heat fluxes to latent heat fluxes, $H/\lambda E$)
derivation (Liu and Yang, 2021). The fundamental conservation equations for states of
moisture and energy over the water surfaces are (Raupach, 2001):
$$\rho c_p \frac{d\theta}{dt} = \frac{H}{h} + \frac{\rho c_p g_e}{h}\left(\theta_e - \theta\right) \tag{3}$$

$$\rho\lambda \frac{dQ}{dt} = \frac{\lambda E}{h} + \frac{\rho\lambda g_e}{h}\left(Q_e - Q\right) \tag{4}$$

where θ (K) is the potential temperature, Q is the specific humidity, $c_p$ (J/kg/K) is the
specific heat capacity of air at constant pressure, $g_e$ (m/s) is the entrainment flux velocity
into the ABL box, and h (m) is the height of the ABL. The subscript e indicates the
variable is evaluated at the upper boundary of the ABL (see Figure 1).
According to Equations (3) and (4), we can obtain a formula to calculate the rate of VPD
(dVPD/dt, see details in Liu and Yang (2021)):
$$\frac{dVPD}{dt} = \frac{\varepsilon_a H - \lambda E}{\rho\lambda h} + \frac{g_e}{h}\Delta_D \tag{5}$$

where $\Delta_D$ is calculated as:
$$\Delta_D = VPD_e - VPD \tag{6}$$

Under the state that air is saturated, the water vapor is continuously transported from the
water surface to the atmosphere, keeping the air saturated. In this case, there is no vertical
moisture gradient, that is, the air near the surface and the air at the upper boundary of the
ABL should be saturated, so VPD and $VPD_e$ are both equal to zero. With Equation (6),
we can know $\Delta_D = 0$.
When air is not saturated, we can rewrite Equation (6) as:

$$\Delta_D = Q - Q_e + \left[ Q_{sat}(\theta_e) - Q_{sat}(\theta) \right] \tag{7}$$

where $Q_e$ is much smaller than Q, and $Q_{sat}(\theta_e)-Q_{sat}(\theta)$ is small (one order of magnitude
smaller than Q), so the $\Delta_D$ roughly equals Q (Raupach, 2001; Liu and Yang, 2021).
Under a relatively long-term (monthly and/or longer), there is a potential VPD budget
(dVPD/dt = 0) over water surfaces (Raupach, 2001), and $g_e$ can be estimated as the
function of H and λE as:

$$g_e = \frac{H + \Lambda \cdot \lambda E}{\rho c_p \gamma_v h} \tag{8}$$

where $\Lambda$ is a constant (0.07), and $\gamma_v$ is the potential virtual temperature gradient in the
free atmosphere above the ABL. $\gamma_v h$ can be set as a fixed value of 7 K (Liu and Yang,
2021). Combining with the VPD budget, Equation (5) and (8), we can obtain the
expression for Bo:

$$Bo = \begin{cases} \dfrac{1}{\varepsilon_a}, \text{equilibrium} \\[2mm] \dfrac{1 - \Lambda\chi}{\varepsilon_a + \chi}, \text{non-equilibrium} \end{cases} \tag{9}$$

where $\chi = \dfrac{\lambda Q}{c_p \gamma_v h}$, a function of Q.

## 2.2 Theoretical formula for α

The surface energy balance is expressed as:

$$R_n = H + \lambda E + G = (1 + Bo)\lambda E + G. \tag{10}$$

Combining Equations (2) and (10), α can be calculated as:

$$\alpha = \frac{1}{1 + Bo} \frac{\varepsilon_a + 1}{\varepsilon_a}. \tag{11}$$

With Equation (9) and (11), we can derive the formula for α:

$$\alpha = \begin{cases} 1, \text{equilibrium} \\[2mm] 1 + \dfrac{(\varepsilon_a \Lambda + 1)\chi}{\varepsilon_a \left[ \varepsilon_a + 1 + (1 - \Lambda)\chi \right]}, \text{non-equilibrium} \end{cases} \tag{12}$$

Equation (12) is one of the main results in this study, and it can estimate α well compared
to a large number of observations (Figure 2, please see the description of observed data
in Section 3).

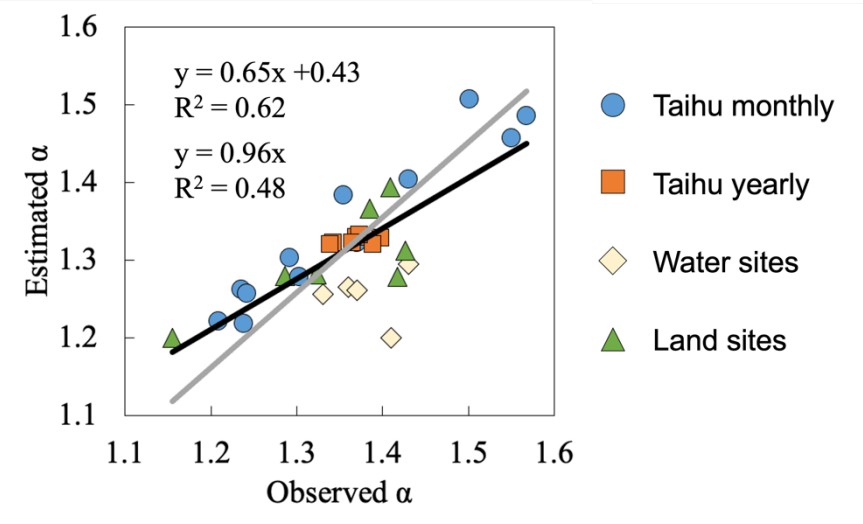


Figure 2. Comparison between observed and Equation (12) calculated α. The black line
is the linear fitting with intercept and the gray line is the linear fitting through origin. The
observed α is inversed by the PT model.
**2.3 The sensitivity of α to air temperature and humidity**
According to the above derivations, we can know that α is not a constant and it changes
with T and Q. The sensitivity of α to T and Q, dα/dT and dα/dQ, determines the variation
of α if the initial α value is given. In this section, we derive explicit equations to estimate
dα/dT and dα/dQ.
Firstly, we decompose α changes in that of T and Q with partial differential equations
based on Equation (11):

$$\frac{\partial \alpha}{\partial T} = -\frac{1}{\left(1+\mathrm{Bo}_{\mathrm{ABL}}\right)^2}\frac{\varepsilon_a+1}{\varepsilon_a}\frac{\partial \mathrm{Bo}_{\mathrm{ABL}}}{\partial T} - \frac{1}{\varepsilon_a^2}\frac{1}{1+\mathrm{Bo}_{\mathrm{ABL}}}\frac{\partial \varepsilon_a}{\partial T},\tag{13}$$

$$\frac{\partial \alpha}{\partial Q} = -\frac{1}{\left(1+\mathrm{Bo}_{\mathrm{ABL}}\right)^2}\frac{\varepsilon_a+1}{\varepsilon_a}\frac{\partial \mathrm{Bo}_{\mathrm{ABL}}}{\partial Q},\tag{14}$$

where partial differential terms of $\dfrac{\partial \mathrm{Bo}_{\mathrm{ABL}}}{\partial T}$ and $\dfrac{\partial \mathrm{Bo}_{\mathrm{ABL}}}{\partial Q}$ can be estimated based on
Equation (9) as:

$$\frac{\partial \mathrm{Bo}_{\mathrm{ABL}}}{\partial T} = -\frac{1-\Lambda\chi}{\left(\varepsilon_a+\chi\right)^2}\frac{\partial \varepsilon_a}{\partial T},\tag{15}$$

$$\frac{\partial \text{Bo}_{\text{ABL}}}{\partial Q} = -\frac{\Lambda \varepsilon_a + 1}{\left(\varepsilon_a + \chi\right)^2} \frac{\partial \chi}{\partial Q}. \tag{16}$$

where terms of $\dfrac{\partial \varepsilon_a}{\partial T}$ and $\dfrac{\partial \chi}{\partial Q}$ can be approximated as:

$$\frac{\partial \varepsilon_a}{\partial T} = \frac{1}{\gamma} \frac{\partial \Delta}{\partial T}, \tag{17}$$

$$\frac{\partial \chi}{\partial Q} = \frac{\lambda}{c_p \gamma_v h}, \tag{18}$$

where $\Delta$ can be calculated as:

$$\Delta = \frac{4098 e_s}{\left(T + 237.3\right)^2}. \tag{19}$$

Combining Equation (13)-(18), we can obtain:

$$\frac{\partial \alpha}{\partial T} = \frac{1}{\gamma} \left[ \frac{1}{\left(1 + \text{Bo}_{\text{ABL}}\right)^2} \frac{1 - \Lambda \chi}{\left(\varepsilon_a + \chi\right)^2} \frac{\varepsilon_a + 1}{\varepsilon_a} - \frac{1}{\varepsilon_a^2} \frac{1}{1 + \text{Bo}_{\text{ABL}}} \right] \frac{\partial \Delta}{\partial T} \tag{20}$$

$$\frac{\partial \alpha}{\partial Q} = \frac{1}{\left(1 + \text{Bo}_{\text{ABL}}\right)^2} \frac{\Lambda \varepsilon_a + 1}{\left(\varepsilon_a + \chi\right)^2} \frac{\varepsilon_a + 1}{\varepsilon_a} \frac{\lambda}{c_p \gamma_v h} \tag{21}$$

We can rewrite the Equation (20) as follows:

$$\frac{\partial \alpha}{\partial T} = -\frac{1}{\gamma} \frac{\chi \left[ \varepsilon_a \left(\Lambda \varepsilon_a + 2\right) + \chi \left(1 - \Lambda\right) + 1\right]}{\left(1 + \text{Bo}_{\text{ABL}}\right)^2 \left(\varepsilon_a + \chi\right)^2 \varepsilon_a^2} \frac{\partial \Delta}{\partial T}, \tag{22}$$

The total differentiation of $\alpha$ is:

$$d\alpha = \frac{\partial \alpha}{\partial T} dT + \frac{\partial \alpha}{\partial Q} dQ, \tag{23}$$

thus $\dfrac{d\alpha}{dT}$ and $\dfrac{d\alpha}{dQ}$ can be written as:

$$\frac{d\alpha}{dT} = \frac{\partial \alpha}{\partial T} + \frac{\partial \alpha}{\partial Q} \frac{dQ}{dT}, \tag{24}$$

$$\frac{d\alpha}{dQ} = \frac{\partial \alpha}{\partial Q} + \frac{\partial \alpha}{\partial T} \frac{dT}{dQ}. \tag{25}$$

With the above equations, we can get theoretical relationships among $\alpha$, T, and Q. This derivation can provide a simple and physically clear estimation for $\alpha$ changes. We also obtained $d\alpha/dT$ and $d\alpha/dQ$ values by fitting measured data using the linear regression

model.
For practical use, we simplified the Equation  (20)  and  (21)  as:
$$\frac{\partial \alpha}{\partial T} = -\frac{1}{\gamma} \frac{\chi}{\varepsilon_a + \chi} \frac{1}{\varepsilon_a^2} \frac{\partial \Delta}{\partial T}$$
(26)

$$\frac{\partial \alpha}{\partial Q} = \frac{\varepsilon_a + 1}{\varepsilon_a (\varepsilon_a + \chi + 1)^2} \frac{\chi}{Q}$$
(27)

We further gave a numerical plot to show how α changes with T and Q (Figure 3). We
plot this figure by setting a $dQ/dT$ gradient from 0.0005, 0.0007, and 0.0009/K to ensure
cover most of the cases over water surfaces. Figure 3 can be used as the lookup graphs to
directly find $d\alpha/dT$  and  $d\alpha/dQ$  values. For example, for a water surface with $dQ/dT$
about 0.0007 /K, the values of $d\alpha/dT$  and  $d\alpha/dQ$  can be found in the second column
of Figure 3.

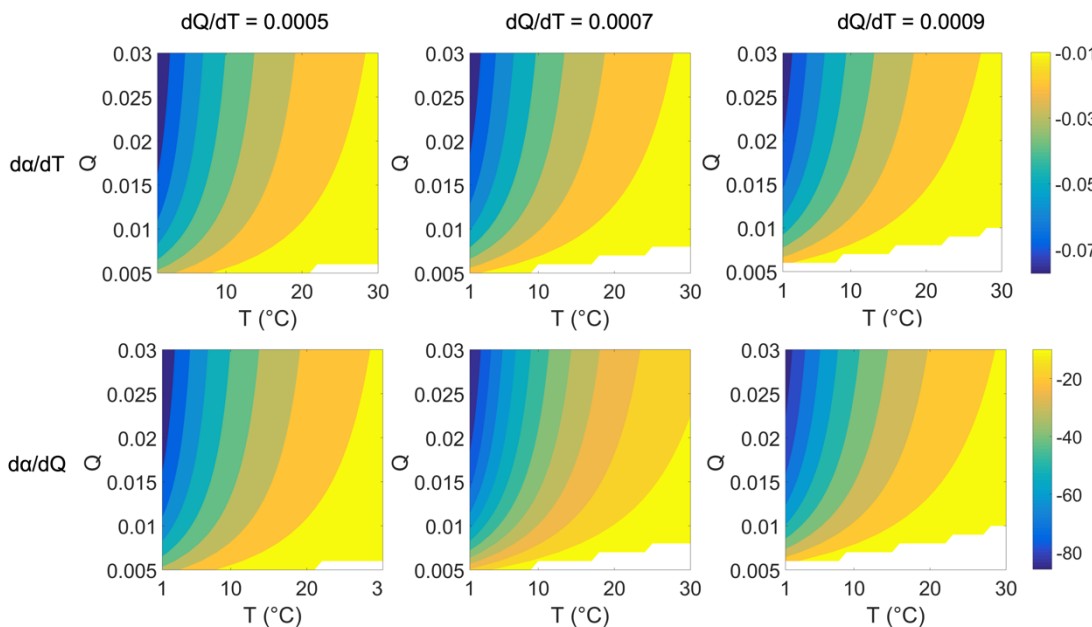


Figure 3. Values of $d\alpha/dT$  and  $d\alpha/dQ$  under different T and Q. The first and second
rows are $d\alpha/dT$  and  $d\alpha/dQ$, respectively. The first to third columns are under different
correlations between Q and T ( $dQ/dT$ ) as 0.0005, 0.0007, and 0.0009/K, respectively.
The blank space in each subpanel refers to values of $d\alpha/dT$  and  $d\alpha/dQ$  are negative,
indicating situations that rarely happen in the real world (i.e., with a very high temperature,
the specific humidity is hardly deficient over wet surfaces).
**3. Cases and applications**
**3.1 Data**
We select data from eddy covariance measurements on several water surfaces (Han and
Guo, 2023): (i) Lake Taihu, located in the Yangtze River Delta, China, with an area of
~2,400 km$^2$, an average depth of 1.9 m (Lee et al., 2014). There are five sites over the
Taihu surface, and the poor-quality data marked with quality flags are removed. (ii) Lake
Poyang, located in the Yangtze Plain, China, with an area of ~3,000 km$^2$ and an average
depth of 8.4 m (Zhao and Liu, 2018). (iii) Erhai, located in the Yun-Gui Plateau of China,
with an area of ~250 km$^2$ and an average depth of 10 m (Du et al., 2018). (iv) Guandu
Ponds, located in Anhui Province, China, with an area of ~0.05 km$^2$ and an average depth
of 0.8 m (Zhao et al., 2019); (v) Lake Suwa, located in Nagano, Japan, with an area of
~13 km$^2$ and an average depth of 4 m (Taoka et al., 2020). Months with negative values
of sensible heat fluxes have not remained. Given the absence of observed heat storage (G)
at some sites, we use the sum of latent heat flux and sensible heat flux (i.e., LE+H) instead
of net radiation minus G ($R_n$-G) as the measure of available energy. Using either LE+H
or $R_n$-G yields identical results, as our objective is to use the available energy to invert
parameter α from observations. The latitude, longitude, and available data period of five
lakes/ponds are listed in Table 1. For α changes in time, we use data from Lake Taihu for
investigation due to its sufficient data length. For α changes in space, we calculate the
average temperature, specific humidity, and α of each lake for comparison.

Table 1. Location and date period of each water body.

| Site | Lat (°) | Lon (°) | Size (km$^2$) | Periods[a] | Sample size (number of months) |
|---|---|---|---|---|---|
| Taihu | 31.23 | 120.11 | 3000 | 2012.01 - 2018.12 | 341[b] |
| Poyang | 29.08 | 116.40 | 2400 | 2013.08 - 2017.09 | 41 |
| Erhai | 25.77 | 100.17 | 250 | 2012.01 - 2018.12 | 24[c] |
| Guandu | 31.97 | 118.25 | 0.05 | 2017.06 - 2019.12 | 31 |
| Suwa | 36.05 | 138.11 | 13 | 2016.01 - 2018.12 | 36 |

Note: a. Periods refer to the date of the first measurement to the date of the last one,
including months for which no data are available. b. There are five eddy covariance
sites over lake Taihu. c. Only climatology monthly data from two periods of 2012-2015
and 2015-2018 are available.
Observations from global flux sites (FluxNet2015 database) are also selected. We first
examine days without water stress based on the following steps (Maes et al., 2019). At
each site, the evaporative fraction (i.e., EF, latent heat flux over the sum of latent and
sensible fluxes) is first calculated, and the days with EF exceeding the 95th percentile EF
and with EF larger than 0.8 remain. Secondly, the days with soil moisture lower than 50%
of the maximum soil moisture (taken as the 98th percentile of the soil moisture series) are
removed. Days having rainfall and negative values of latent and sensible heat fluxes are
also not included. As a result, a total of ~700 non-water-stressed site-days pass the
criterion. Data is divided into seven vegetation types including croplands (CRO),
wetlands (WET), evergreen needleleaf and mixed forests (DNF_MF), evergreen
broadleaf and deciduous broadleaf forests (EBF_DBF), grasslands (GRA), close
shrublands (CSH), and woody savanna (WSA), to analyze α changes in space. It should
be noted that we do not average the daily data to a monthly scale due to variations in data
sizes across different months for a specific site. Instead, we organize the selected daily
data by vegetation types, as the primary objective of utilizing land fluxes data is to assess
the derived relationship spatially rather than temporally.
We also collect ocean surface data from 11 CMIP6 models (under scenario SSP585, Table
2) from 2021-2100 to see the temporal changes in α. The calculation is limited to the
latitudinal range 60°S to 60°N, and takes all ocean surface grids as a whole (Roderick et
al., 2014). We average the monthly data to the yearly scale and calculate α every ten years
from 2021 to 2100 (i.e., 2021-2030, 2031-2040, etc.).

Table 2. CMIP6 models used in this study.

| Model | Nation | Institute |
| --- | --- | --- |
| ACCESS-ESM1-5 | Australia | CSIRO |
| CanESM5 | Canada | CCCma |
| CESM2-WACCM | USA | NCAR |
| CMCC-CM2-SR5 | Italy | CMCC |
| CMCC-ESM2 | Italy | CMCC |
| FGOALS-g3 | China | CAS |
| FIO-ESM-2-0 | China | CAS |
| MPI-ESM1-2-HR | Germany | MPI-M |
| MPI-ESM1-2-LR | Germany | MPI-M |
| NorESM2-LM | Norway | NCC |
| NorESM2-MM | Norway | NCC |

Note: CSIRO: Commonwealth Scientific and Industrial Research Organization;
CCCma: Canadian Centre for Climate Modelling and Analysis; NCAR: National Center
for Atmospheric Research; CMCC: Euro-Mediterranean Center on Climate Change;
CAS: Chinese Academy of Sciences; MPI-M: Max Planck Institute for Meteorology;
NCC: Norwegian Climate Centre.
**3.2 Results**
**(1) Temporal and spatial changes in α**
We used yearly and climatology monthly (from Jan to Dec) data from Lake Taihu to
investigate the temporal variation in α. α is firstly inversed by the PT model and
measurements, and then we found significant negative relationships of α with both T and
Q (Figure 4). On the yearly scale, the regressed values of dα/dT and dα/dQ are -
0.029/°C and -47.42, and the values on the seasonal scale are -0.014/°C and -20.75,
respectively. dα/dT on the seasonal scale is higher than that on the yearly scale because
the variation range of α on the seasonal scale is more extensive. Theoretical derivations
can roughly reproduce the sensitivity of α to T and Q, although there is some potential
uncertainty from interannual variations (Table 3). We also analyzed the results on the ten-
day scale and obtained similar findings (see Appendix Figure A1 and Table A1).

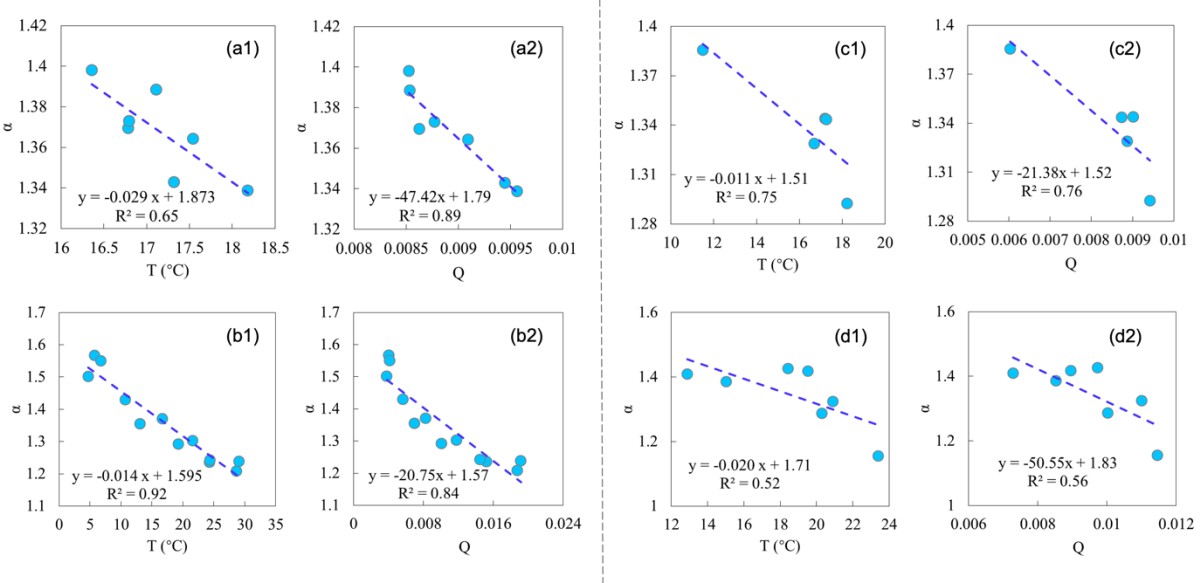


Figure 4. Temporal and spatial relationships of α and temperature (T) and specific
humidity (Q). (a-b) Temporal relationships based on Lake Taihu data: (a) yearly data, and
(b) climatology monthly data. (c-d) Spatial relationships: (c) data from five water surface
sites, and (d) land surface data from FluxNet2015, each circle representing one vegetation
type. The linear regression line and correlation coefficient ($R^2$) are shown in each
subpanel.

Table 3 Sensitivity of α to temperature (T) and specific humidity (Q) by regression and
theoretical derivation.

|          |             | dα/dT (/°C) | | dα/dQ | |
|----------|-------------|------------|------------|------------|------------|
|          |             | regression | derivation | regression | derivation |
| Temporal | yearly      | -0.029     | -0.016     | -47.42     | -20.33     |
|          | seasonally  | -0.014     | -0.011     | -20.75     | -18.38     |
| Spatial  | water sites | -0.011     | -0.009     | -21.38     | -12.22     |
|          | land sites  | -0.020     | -0.013     | -50.55     | -31.97     |


Spatial relationships of α with T and Q are similar to that in time, i.e., higher T and Q
generally correspond to lower α, supported by measurements over both water and land
surfaces (Figure 4). For the water surfaces, the values of $d\alpha/dT$ and $d\alpha/dQ$ are -
0.011/°C and -21.38, and the values for land surfaces are -0.020/°C and -50.55. The
derived $d\alpha/dT$ and $d\alpha/dQ$ matched roughly well with the regressed values, despite
more or less errors (Table 3). The correlations (represented by $R^2$ in Figure 4) between α
and T, α and Q of water surfaces are higher than those over the land surfaces. This
indicates that changes in α are more associated with T and Q over water surfaces, which
may be because T and Q dominate the water surface evaporation process, while some
other factors, like vegetation and wind speed, also play specific roles over land surfaces.
Based on Equation (20) to (22), $\partial\alpha/\partial T$ is always a negative value, and $\partial\alpha/\partial Q$ is
always positive. The regressed and derived $d\alpha/dT$ and $d\alpha/dQ$ are both negative.
Combined with Equations (24), (25) and the positive relationship between T and Q, the
$\partial\alpha/\partial T$ plays a more critical role in determining (the signs of) $d\alpha/dT$ and $d\alpha/dQ$, that
is, $\left|\partial\alpha/\partial T\right| > \partial\alpha/\partial Q \cdot dQ/dT$ and $\left|\partial\alpha/\partial T \cdot dT/dQ\right| > \partial\alpha/\partial Q$. Specifically, based on the data
from lake Taihu (for detecting α changes in time) and data from different water surface
sites and land surface sites (for detecting α changes in space), we found the contribution
of $\partial\alpha/\partial T \cdot dT$ to $d\alpha$ is ~70%, much more significant than that of $\partial\alpha/\partial Q \cdot dQ$ of ~30%
(Table 4). Therefore, according to the evaporation process over the wet surface (Section
2.1) and the above analyses, we can conclude that α is fundamentally controlled by T and
modulated by Q.
Table 4. Contributions of changes in temperature (T) and specific humidity (Q) to
changes in α.

|  |  | $d\alpha$ | contribution of $\dfrac{\partial\alpha}{\partial T}dT$ | contribution of $\dfrac{\partial\alpha}{\partial Q}dQ$ |
|---|---|---|---|---|
| Temporal | yearly | -0.029 | 70% | 30% |
|  | seasonally | -0.256 | 66% | 34% |
| Spatial | water sites | -0.080 | 70% | 30% |
|  | land sites | -0.136 | 74% | 26% |
| Average |  | ---- | 70% | 30% |

Note: Since $d\alpha = \dfrac{\partial\alpha}{\partial T}dT + \dfrac{\partial\alpha}{\partial Q}dQ$, the contribution of $\dfrac{\partial\alpha}{\partial T}dT$ is calculated as
$\left|\dfrac{\partial\alpha}{\partial T}dT\right| \Big/ \left|\dfrac{\partial\alpha}{\partial T}dT + \dfrac{\partial\alpha}{\partial Q}dQ\right|$, and is the contribution of $\dfrac{\partial\alpha}{\partial Q}dQ$ calculated as
$\left|\dfrac{\partial\alpha}{\partial Q}dQ\right| \Big/ \left|\dfrac{\partial\alpha}{\partial T}dT + \dfrac{\partial\alpha}{\partial Q}dQ\right|$. $d\alpha$ refers to the estimated variation of α from lowest to highest
T (also from lowest to highest Q since T and Q are generally positively correlated).
Derived $d\alpha/dT$ and $d\alpha/dQ$ have more or less errors compared to the regressed values.
Several reasons can explain this: (i) errors in measurements of eddy covariance systems;
(ii) the additional factors other than T and Q, like wind speed, can also influence $\alpha$; (iii)
the relationship of $\alpha$ and T (also $\alpha$ and Q) cannot be well represented by the linear
regression model. Besides, the water surface size effects on evaporation and $\alpha$, reported
by Han and Guo (2023), are not well considered in the presented derivation. Nevertheless,
the derived expression can fairly match the observations of water bodies with various
sizes (Table 3).

### (2) Potential applications for global projections

Based on CMIP6 ocean surface data, we also detected significant negative relationships
of $\alpha$ with T and Q (Figure 5). $d\alpha/dT$ and $d\alpha/dQ$ obtained by the linear regression are -
0.009/°C and -11.54, respectively. The derived $d\alpha/dT$ and $d\alpha/dQ$ are close to the
regressed value as -0.009/°C and -10.74. We further compared the changes in T, Q, and
heat fluxes between the first and the last ten years in 2021-2100 (Table 5). To the end of
this century, CMIP6 models predict that ocean average available energy ($R_n$-G) and latent
heat flux (also evaporation) will increase by ~3.1 W/m$^2$ and ~6.0 W/m$^2$, respectively.
Using the PT model with the fixed $\alpha$ (1.26), predicted evaporation shows an increase of
~8.0 W/m$^2$, far higher than climate models' direct output (with a relative bias of ~30%).
Based on derived $\alpha$, ocean evaporation shows a much smaller increase of ~5.8 W/m$^2$, with
less than 5% relative bias compared to CMIP6 values (Figure 6). This indicates that
changes in $\alpha$ should be well considered for the long-term projections. So here we suggest
introducing the negative relationship between $\alpha$ and T, proposed in this study, into the
original PT model to correct for the overestimated sensitivity of evaporation to
temperature (Liu et al., 2022), which could also improve the reliability of global long-
term drought predictions (Greve et al., 2019).

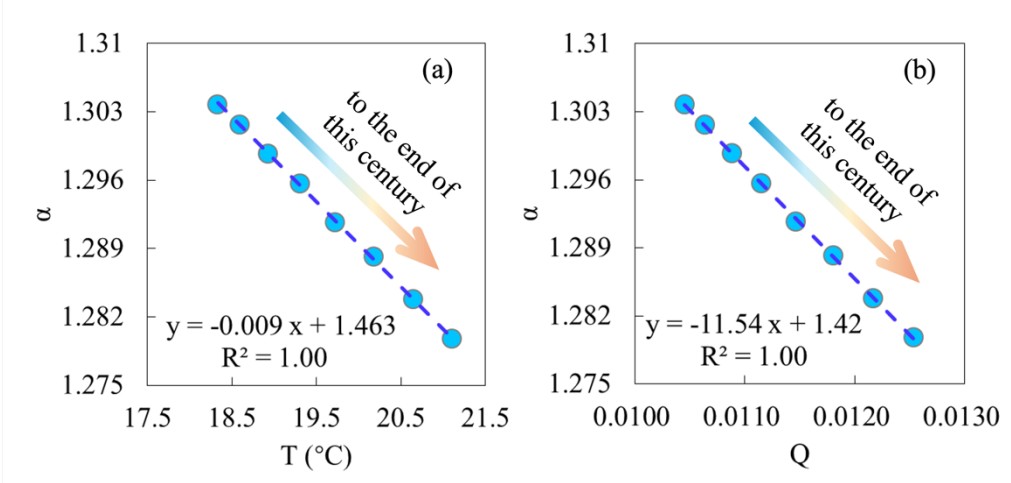


Figure 5. Temporal relationship of (a) $\alpha$ and temperature (T), and (b) $\alpha$ and specific

humidity (Q) over global ocean surfaces. Each dot denotes the data in each 10-year window (2021-2030, 2031-2041, …, 2091-2100), from left to right is from 2021-2030 to 2091-2100.

Table 5. Ocean surface temperature, specific humidity, and heat fluxes at the first ten years (2021-2030) and the end of the $21^{st}$ century (2091-2100). T, Q, $R_n$-G, and LE are direct outputs of climate models. α-CMIP refers to α inversed by the PT model with CMIP data. $LE_{PT}$ is calculated by the PT model with fixed α at 1.26. α-ABL refers to α estimated by the ABL model. $LE_{ABL}$ is calculated by the PT model with α-ABL.

| Period | T | Q | $R_n$-G | LE | α-CMIP | $LE_{PT}$ | α-ABL | $LE_{ABL}$ |
|---|---|---|---|---|---|---|---|---|
| | (°C) | (-) | (W/m$^2$) | (W/m$^2$) | | (W/m$^2$) | | (W/m$^2$) |
| 2021-2030 | 18.1 | 0.010 | 122.9 | 106.8 | 1.304 | 103.2 | 1.316 | 107.7 |
| 2091-2100 | 21.1 | 0.013 | 126.0 | 112.9 | 1.279 | 111.2 | 1.287 | 113.5 |
| Δ | 3.0 | 0.003 | 3.1 | 6.1 | -0.025 | 8.0 | -0.029 | 5.8 |

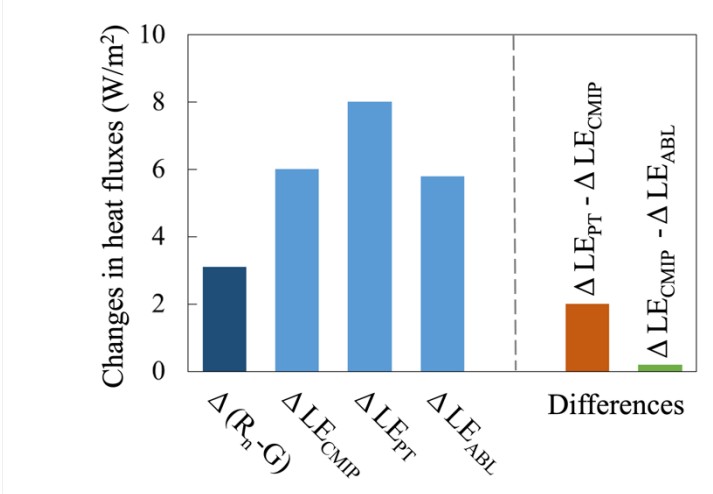

Figure 6. Stylized diagram showing the average changes in heat fluxes over global ocean surfaces.

## 4. Discussions and Conclusions

In this study, we employed an open boundary layer model with a governing potential VPD budget (Raupach, 2001, 2000), originally integrated by Liu and Yang (2021), to formulate an expression for the Priestley-Taylor coefficient, α. Notably, the governing equation allows the derived expression to have no calibrated parameters and can estimate a precise α value with normal observations, rendering it superior to other methods that are also built with the boundary layer theory (Lhomme, 1997b; Van Heerwaarden et al., 2009). With the expression and a variety of measurements, we further demonstrated that temperature exerts a more significant influence on variations in α, as opposed to specific humidity. We suggest that for studies focusing on evaporation and/or drought projections, it is crucial

to thoroughly characterize the negative correlation between α and temperature, a
relationship easily determined using the derived expression.
It should be noted that except for the PT model, the PM-based model can be also used to
estimate wet surface evaporation (Penman, 1948; Shuttleworth, 1993). While PM-based
equations encapsulate all processes that possibly affect evaporation, the PT model, taking
evaporation as a simple function of radiation and temperature, takes more account of the
feedback/balance between the surface and near atmosphere (Figure 1). Besides, it has
been noted that the PM-based models may fail at certain limits, and cannot capture the
sensitivity of evaporation to temperature changes (Liu et al., 2022; McColl, 2020). So in
this case, also with the fact that the PT model is currently one of the most popular
equations due to its low input requirements, revisiting this classic model can greatly
promote its adaption under the changing climate. Meanwhile, some revised PT equations
can also be used to estimate the parameter α (Yang and Roderick, 2019; De Bruin and
Holtslag, 1982). However, these modifications often exhibit significant deviations
(Figure A2). Specifically, the model developed by De Bruin and Holtslag (1982) is based
on data from one specific site in the Netherlands, and the model built by Yang and
Roderick (2019) comes from the fitness of global ocean surface data. These equations are
primarily calibrated to match observed evaporation rates, while the underlying process is
generally overlooked.
In Section 2.1, it was suggested that $\Delta_D = 0$ for the saturated air while $\Delta_D \approx Q$ for the
non-saturated air. In theory, it is expected that the transition track between saturated and
non-saturated states should be continuous and smooth. That is, the changes in the value
of $\Delta_D$ between the saturated (0) and non-saturated (Q) states should follow the variations
in air energy and moisture (Figure 7). Since the relative humidity (RH) includes both
information on air temperature and humidity, here we introduce a possible track of $\Delta_D$
depending on RH as: $\Delta_D = \psi(RH) \cdot Q$. As we expect, the value of $\Delta_D$ approaches 0 when
the air is very moist (i.e., very close to the saturated state and RH close to 1), so $\psi$ should
be a nonlinear and monotone convex function of RH. We give a possible expression of
$\psi(RH)$ as:

$$\psi(RH) = 1 - \frac{1}{1 + m \times \left(\dfrac{RH_{max} - RH}{RH - RH_{min}}\right)^n} \tag{28}$$

where $RH_{max}$ is 1, and $RH_{min}$ is 0.6 (Mccoll and Tang, 2023) over the water surfaces. m
and n are shape parameters. To make $\psi(RH)$ simple, we fixed n at 1, and let m be 100.
The relationship between $\psi(RH)$ and RH can be viewed in Figure 7 (b). For a specific
case that T at 18 °C, we show the changes in Bo and α with RH in Figure 7 (c)-(d).
Although there is a dramatic shift in Bo or α, it appears when RH is at 0.95-1, which is
outside the vast majority of actual cases (RH is generally smaller than 0.9 on a monthly
or longer scale). After the shift point, with RH decreases, $\psi(\text{RH})$, Bo, and α remain
roughly stable. It is worth noting that Equation (28) (with specific parameters) is one
possible case that connects the transition between saturated and non-saturated air states,
a fine determination may be affected by local conditions, but $\Delta_D$ value around Q is
expected for most of the cases.

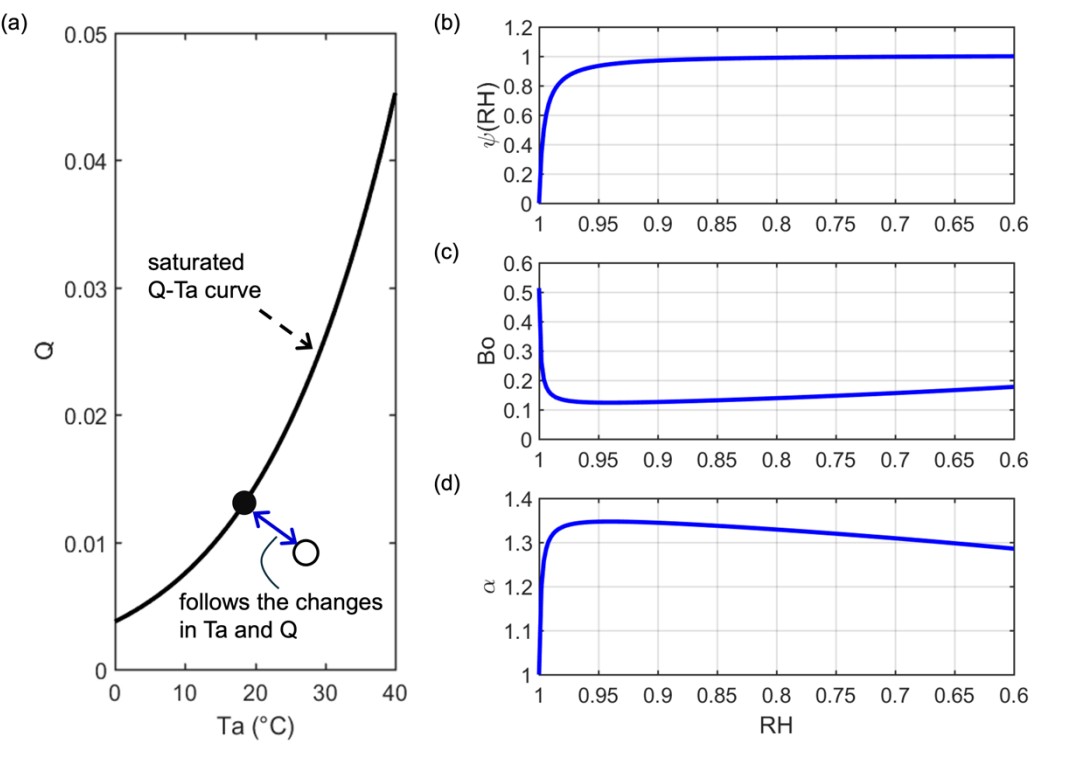


Figure 7. (a) Transition between saturated and non-saturated air states. The filled circle
represents one case in which the air is saturated (saturated state) and the open circle
represents one case in which air is not saturated (non-saturated state). (b) Relationship
between $\psi(\text{RH})$ and RH with Equation (28). (c)-(d) Changes in Bo and α as the
function of RH when air temperature is fixed at 18 °C.
We recommend utilizing the derived model under warm conditions, for example, when
the air temperature exceeds zero, to account for the prerequisite of a well-mixed boundary
layer. In extremely cold regions or seasons, the water surface temperature can be lower
than the air temperature, resulting in a downward sensible heat flux (De Bruin, 1982).
Under such circumstances, the boundary layers exhibit relative stability and may not
reach a well-mixed state. Additionally, we advise adopting a temporal scale ranging from
weekly to monthly when applying the derived model. This is because the potential VPD
budget (the governing equation) may not be rapidly achieved, such as on a diurnal or daily
basis. Furthermore, over a longer term, the sensible heat flux typically manifests as
upward in the majority of scenarios than on a fine temporal scale.
The derived formula for α has important practical meanings. For example, it would be
useful for estimating water surface evaporation and actual evapotranspiration based on
the PT model (Miralles et al., 2011; Maes et al., 2019). It can also help to constrain the
relationships among α, T, and Q in the complementary relationship, whose performance
previously depended on the inversed α (Liu et al., 2016). Besides, considering the impacts
of changing climate on α can significantly improve the performance of the hydrologic
model in runoff simulations and predictions (Pimentel et al., 2023).

**Appendix A.**

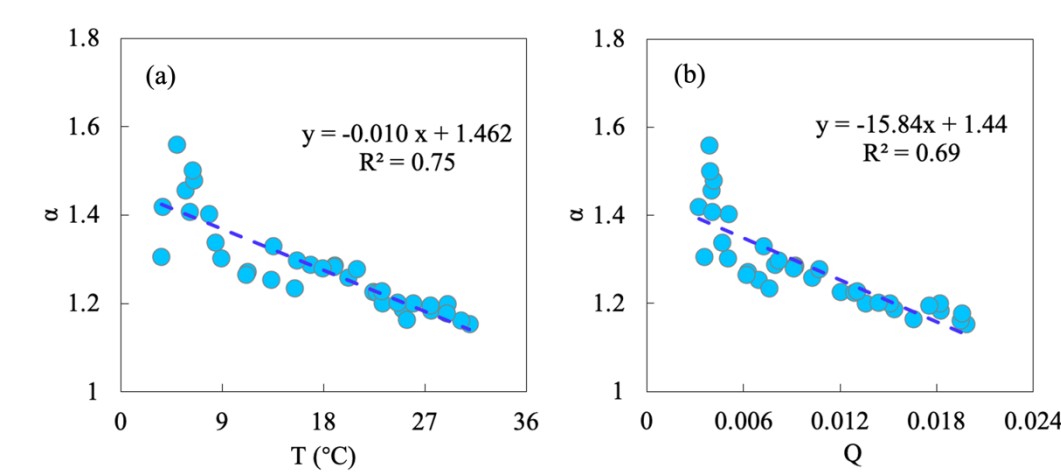

Figure A1. Relationships of $\alpha$ with (a) temperature (T) and with (b) specific humidity (Q) on the ten-day scale using water surface observations collected over Lake Taihu.

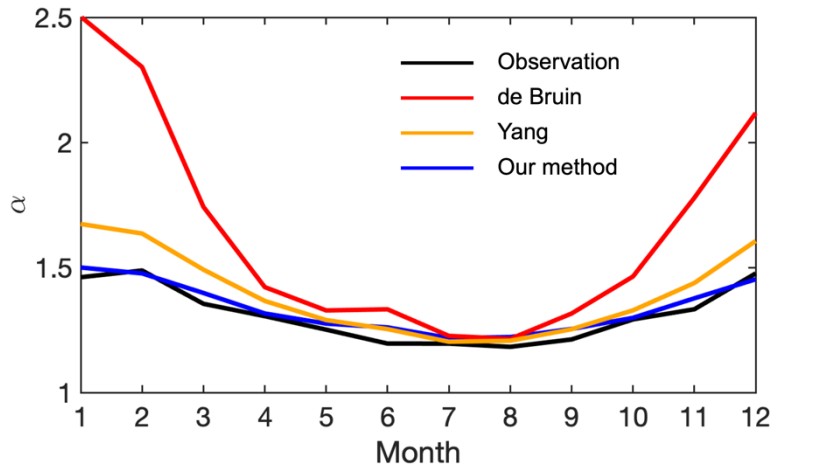

Figure A2. Observed (black) and estimated $\alpha$ over lake Taihu. The blue line is $\alpha$ estimated with our method, and the red and orange lines are with two revised PT

equations. The red line represents $\alpha=1+\dfrac{20}{\frac{\Delta}{\Delta+\gamma}(R_n-G)}$ (De Bruin and Holtslag, 1982), and

the orange line represents $\alpha=\dfrac{\Delta+\gamma}{\Delta+0.24\gamma}$ (Yang and Roderick, 2019).

Table A1. Sensitivity of $\alpha$ to temperature (T) and specific humidity (Q) on the ten-day scale.

| d$\alpha$/dT (/°C) | | d$\alpha$/dQ | |
| --- | --- | --- | --- |
| regression | derivation | regression | derivation |
| -0.010 | -0.011 | -15.84 | -18.12 |

**Author Contributions**

Conceptualization: Ziwei Liu, Hanbo Yang. Data curation: Ziwei Liu. Formal analysis: Ziwei Li. Funding acquisition: Hanbo Yang. Methodology: Ziwei Liu, Hanbo Yang. Software: Ziwei Liu. Supervision: Hanbo Yang. Writing – original draft: Ziwei Liu. Writing – review & editing: Changming Li, Taihua Wang, Hanbo Yang.

**Data availability**

Data of Lake Taihu can be obtained from Harvard Dataverse, https://doi.org/10.7910/DVN/HEWCWM. The data of Poyang Lake can be obtained from Zhao and Liu (2018) and Gan and Liu (2020). The data of Erhai can be obtained from Du et al. (2018). The data of Guandu can be obtained from Zhao et al. (2019). The data of Suwa lake can be obtained from the AsiaFlux (http://asiaflux.net/index.php?page_id=1355). FluxNet 2015 data are available at https://fluxnet.fluxdata.org/data/download-data/. CMIP6 data can be obtained from Earth System Grid Federation (https://esgf-node.llnl.gov).

**Acknowledgments**

This study is financially supported by the National Natural Science Foundation of China (grant nos. 51979140, 42041004).

**Competing interests**

There are no competing interests.

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
