# Peer review of "Estimating the sensitivity of the Priestley-Taylor coefficient to air"

_Hydrology and Earth System Sciences, 2024_

## Author Comment (AC1)

We greatly appreciate the reviewer for providing valuable and constructive comments on our manuscript. Each comment has been thoroughly considered. In the following section, the original comments are presented in black, and our responses are provided in blue. To facilitate navigation, codes such as C1 (comment 1) and C2 (comment 2) have been included. As per the standard procedure of the journal, we are presenting only our replies to the reviewer in this round, without including the revised manuscript.

C1: The manuscript provides a theoretical derivation of the PT coefficient and evaluates the results by using wet-surface measurements. The manuscript provides some in-depth understanding on the variation of PT coefficient and shows that the value is also essential for hydrological simulation and projections. The manuscript is written in a organized structure and the contents is supported by in situ measurements. However, there are still one major issue need to be clarified.

Response: Thanks for your positive evaluation of our work. Please see our reply below.

C2: We all know that the Priestley-Taylor model has limitations in its application. The PT model should be applied at the appropriate temporal resolution, however, such an issue has been ignored in the present manuscript. In Table 3 and Figure 4, the results show significant differences in seasonal data and yearly data. Thus, the obtained values should also be different at different temporal resolutions. Thus, it would be nice to understand whether the derived relationships also fit at the temporal resolution of weeks or ten-days, or at least to indicate that the results are reasonable at ?? temporal resolution.

Response: Thank you for your comment. We have applied the derived expression on a ten-day scale using measurements collected over Lake Taihu. The data has been averaged for each ten-day period (1$^{st}$ to 10$^{th}$ day, 11$^{th}$ to 20$^{th}$ day, …, 351$^{st}$ to 360$^{th}$ day) within a year. Subsequently, we have averaged the data across all years and sites to establish a climatology dataset at the ten-day scale, resulting in a sample size of 36 (excluding the last five or six days of the year).  The observations reveal that the values of dα/dT and dα/dQ on the ten days are -0.010 /°C and -15.84, respectively (refer to Figure R1). The derived values of dα/dT and dα/dQ are -0.011 /°C and -18.12, respectively, demonstrating close alignment with the observations. This consistency indicates that the derived relationships hold validity across a broad range of temporal scales, from ten-day to annual. The detailed results have been included in the Appendix.

[Figure]

Figure R1. Relationships of $\alpha$ and (a) temperature (T) and (b) specific humidity (Q) on the ten-day scale using water surface observations collected over Lake Taihu. The value of $d\alpha/dT$ or $d\alpha/dQ$ is represented by the slope of the linear regression (dashed line).

C3: Further, the authors select global flux sites data in the evaluation and the days with soil moisture lower than 50% of the maximum soil moisture are removed. Then, how to obtain monthly data at these flux sites. Some details should be given to clarify this issue.

Response: Apologies for any confusion in the text. Initially, we identify non-water-limit site-days based on the criteria outlined in the main text. Subsequently, we do not average the daily data to a monthly scale due to variations in data sizes across different months for a specific site. Instead, we organize the selected daily data by vegetation types, as the primary objective of utilizing land fluxes data is to assess the derived relationship spatially rather than temporally. These specifics have been elucidated in the revised manuscript.

C4: In the equations, all the variables have no units in the manuscript and the abbreviations of CSIRO also have no full names. I suggest the author to include a table to include the unites of each variable and to show full names of the abbreviations in the appendix.

Response: Thanks for your comment. We have now included the units for each variable at their initial appearance in the manuscript. Additionally, the complete names of the institutions associated with the CMIP6 models have been provided in the appendix for clarity and reference.

---

## Author Comment (AC2)

We greatly appreciate the reviewer for providing valuable and constructive comments on our manuscript. Each comment has been thoroughly considered. In the following section, the original comments are presented in black, and our responses are provided in blue. To facilitate navigation, codes such as C1 (comment 1) and C2 (comment 2) have been included. As per the standard procedure of the journal, we are presenting only our replies to the reviewer in this round, without including the revised manuscript.

C1: This is a paper that tries to understand the Priestley-Taylor parameter alpha based on theoretical derivations from a coupled mixed-layer model. I find this work very interesting, but at the same time find that it does not link well enough to earlier work on the topic and therefore I recommend the editor to reject the paper until this has been repaired.

Response: Thank you for evaluating our research and finding it interesting. We would like to emphasize that the main objective of this study is to offer a straightforward and physically grounded approach for determining the coefficient alpha and examining its variations with air temperature and humidity levels. While previous studies (including all papers you mentioned) have demonstrated that alpha is a variable rather than a constant parameter, the hydrology field predominantly employs a fixed value of alpha = 1.26, despite these earlier findings being over two decades old. This prevailing practice persists due to the absence of a clear, simple, and compelling method for estimating alpha. Our research addresses this by introducing a Bowen ratio estimation based on boundary layer theory to establish an expression for alpha. The key enhancements of our method lie in its parameter-free nature and its ability to accurately estimate alpha using standard air temperature and humidity measurements. While some studies you mentioned below proposed alpha calculation methods, these approaches are often constrained by the complexity of determining specific parameters and then they primarily rely on **numerical simulations** that try to give all cases of alpha, rather than actual field measurements. In our study, we not only streamline alpha calculation but also validate our proposed expression using a diverse set of observations to demonstrate its robustness in assessing alpha responses to changing environmental conditions. Our findings suggest that future research applying the PT model to estimate evaporation and its variations can readily utilize our expression without additional complexity. Given the unique scientific contribution of our work in systematically analyzing this aspect, we believe that our research will be of significant benefit to our community, particularly within the field of hydrology.

C2: The work in this paper is far from new in my view, and previous papers have presented the matter in more detail than this paper. I would like to ask the authors to make a detailed comparison to the math in the papers mentioned below, and explain where they deviate and where they are similar. Also I would like to ask them which are the actual new insights emerging from their work that the previous papers could not provide.

Response: We appreciate your valuable feedback. In our research, we have introduced a novel approach to estimating the Bowen ratio and alpha. Our method stands out for its simplicity and lack of additional parameters, offering a clear and physics-based solution. By utilizing observational data within a partially open system framework and integrating a potential vapor pressure deficit budget, we have achieved a robust and reliable estimation. This unique feature of our approach allows us to determine precise values for alpha and its sensitivity to climate conditions based on actual observations, without introducing uncertainties from additional parameters. Setting our approach apart from previous methodologies that relied on numerous parameters or empirical limitations, we have addressed these constraints effectively, enhancing the practicality and utility of our method in diverse research and application scenarios.

C3: The papers of Raupach (BLM, 2000, QJ 2001) present an very in-depth analysis of equilibrium evaporation in a partially open system such as an entraining convective boundary layer. These are in my view the best papers written on this topic and should be studied in much more detail by the authors.

Response: Thanks for your suggestions. Yes, we wholeheartedly agree that these two papers represent exemplary studies on this topic. In fact, the paper where we introduced the model for estimating the Bowen ratio draws heavily from the insights provided in these two seminal works (Liu et al., 2022). We consistently reference these papers in all our related research, acknowledging their foundational contributions to our field of study. It is important to clarify that our focus (the improvement) is on providing a clear expression for the Bowen ratio, supported by a wealth of observational data. This aspect is fundamental to this study and has been extensively discussed in our previous publication (Liu et al., 2022).

C4: Also, the paper of LHomme, 1997, BLM present an alternative derivation for alpha and should be considered in detail as well.

Response: Thanks. Their paper utilized the PM model in conjunction with a boundary layer model to estimate the parameter alpha. One key limitation

identified in their study is using a closed system boundary layer model, which hinders the proper consideration of entrainment flux effects and consistently yields an alpha value near 1 over water surfaces. In contrast, as you previously mentioned, our fundamental Bowen ratio estimation operates within a partially open system, offering a more physically robust framework. Another notable concern regarding LHomme's paper arises from the inversion of alpha using the PM model, with reported shortcomings in capturing temperature sensitivity and potential failures in certain limiting cases. Additionally, employing the PM model introduces additional parameters such as surface resistance and aerodynamic resistance, thereby increasing computational complexity and introducing additional uncertainties to some extent.

C5: Then, the paper of van Heerwaarden et al., (QJ, 2009) presents a full derivation of the Priestley-Taylor parameter alpha, from a mixed-layer model perspective and hence does exactly that what the authors of this paper intend to do. However, this paper does not make any detailed comparison. I would like to learn where similarities are found and where differences arise.

Response: Thanks for your insights. Both their paper and this study utilized the boundary layer model to estimate alpha. We employed a similar conservation equation to characterize the energy and moisture states in the box model. The distinguishing factor between their work and ours is that they did not incorporate the potential vapor pressure deficit budget as the governing equation for solving the Bowen ratio expression. Consequently, their expression includes a series of parameters. They conducted numerical experiments across a wide range of these parameters to delineate a domain encompassing possible variations in alpha. In contrast, our study integrates the potential vapor pressure deficit budget, enabling us to derive an accurate value for the Bowen ratio based on actual air temperature and humidity levels, without the need for additional parameters. It is important to note that the applicability of the budget stems from the temporal scale under consideration, which ranges from ten days to monthly—longer than the hourly or diurnal scale where budget constraints may not be achieved. Therefore, we acknowledge their work offers an efficient method for estimating alpha on a finer time scale, but our approach provides a more parameter-free and simple estimation based on specific observations.

C6: Concerning the contents, the authors depart from the earlier papers in calling the non-saturated state non-equilibrium, while the previous mentioned papers show

that a non-saturated equilibrium exist for open systems. This might require some extra discussion and maybe some rethinking of the chosen definitions.

Response: Apologies for the lack of clarity. In the context of our study, 'equilibrium state' denotes saturated air, while 'non-equilibrium' signifies non-saturated air. To prevent any confusion, we have updated the terminology in the revised manuscript to explicitly use "saturated/non-saturated air" instead of "equilibrium/non-equilibrium state".

---

## Author Response (AR2)

Dear Dr. Su,

We greatly appreciate your providing fast, positive, and valuable comments on our manuscript. We have addressed your concerns in the revised manuscript.

Comment 1: Add a sentence discussing the role of G, as you have responded 'Concerning the issue of G, we do not compute it but utilize the sum of LE and H (as opposed to Rn minus G) from observations as the available energy, since our goal is to estimate the alpha parameter, so using either LE+H or Rn-G will yield identical results.' in the same fashion.

Response: Thanks. We have added a sentence "Given the absence of observed heat storage (G) at some sites, we use the sum of latent heat flux and sensible heat flux (i.e., LE+H) instead of net radiation minus G ($R_n$-G) as the measure of available energy. Using either LE+H or $R_n$-G yields identical results, as our objective is to use the available energy to invert parameter α from observations" in the Section of Data Introduction.

Comment 2: It would be useful to present your Fig. R2 in the discussion part.

Response: Thanks, done. In the discussion part, we have added a sentence "Meanwhile, some revised PT equations can also be used to estimate the parameter α (Yang and Roderick, 2019; De Bruin and Holtslag, 1982). However, these modifications often exhibit significant deviations (Figure A2). Specifically, the model developed by De Bruin and Holtslag (1982) is based on data from one specific site in the Netherlands, and the model built by Yang and Roderick (2019) comes from the fitness of global ocean surface data. These equations are primarily calibrated to match observed evaporation rates, while the underlying process is generally overlooked."

In the revised manuscript, we have embedded the appendix within the main text (due to only two figures and one table included), eliminating the supplementary file in the latest version.

Thanks again for overseeing the review process. Should you require further information or clarification, please feel free to contact me.

Yours sincerely,

Hanbo Yang (on behalf of the author team)
Tsinghua University
Email: yanghanbo@tsinghua.edu.cn